# The Impact of Self-Regulation Education Combined with Continuous Glucose Monitoring (CGM) on Diabetes Outcomes: A Randomized Controlled Study

**DOI:** 10.3390/nursrep15030094

**Published:** 2025-03-12

**Authors:** Hsiao-Yun Chang, Kuei-Chun Yeh, Yu-Yao Huang, Jui-Hsiang Li

**Affiliations:** 1Department of Nursing, Chang Gung University of Science and Technology, Taoyuan 33303, Taiwan; 2Division of Endocrinology and Metabolism, Department of Internal Medicine, Linkou Chang Gung Memorial Hospital, Taoyuan 33305, Taiwan; yyh@cgmh.org.tw; 3Taoyuan General Hospital, Taoyuan 33304, Taiwan; awesome4610546@gmail.com (K.-C.Y.); li631001@gmail.com (J.-H.L.); 4College of Medicine, Chang Gung University, Taoyuan 33302, Taiwan

**Keywords:** continuous glucose monitoring (CGM), diabetes, self-efficacy, health behaviors, glucose control

## Abstract

In previous studies exploring continuous glucose monitoring (CGM), there has been a limited focus on how CGM influences key behavioral outcomes such as self-efficacy, health behaviors, and medication adherence. **Background/Objectives**: The aim of this study was to assess the impact of combining self-regulation health education with CGM on medication adherence, diabetes self-efficacy, diabetes health behaviors, and glucose control in individuals with diabetes. **Methods**: A randomized controlled study, reported following the CONSORT 2010 reporting guidelines. Individuals with diabetes volunteered to participate and were randomly allocated into two groups: the CGM group (n = 34) and the control group (n = 34). The CGM group received real-time CGM devices and education on self-regulation theory to enable them to self-adjust health promotion strategies and behaviors, while the control group received routine diabetes health education focusing on self-monitoring of blood glucose. Outcome measures included medication adherence, diabetes self-efficacy, diabetes health behaviors, and glucose control. **Results**: The CGM group demonstrated consistent diabetes self-efficacy, significant improvements in diabetes health behaviors, and a reduction in HbA1c levels over time. However, no significant differences in outcomes were observed between the CGM group and the control group. **Conclusions**: The use of continuous glucose monitoring (CGM) provides continuous, real-time glucose data. When combined with self-regulation education, it may help provide personalized insights into how specific foods, activities, medications, and stress levels affect blood glucose levels. This allows individuals with diabetes to make personalized adjustments to their lifestyle and treatment plans to optimize their blood sugar control.

## 1. Introduction

Diabetes management presents a multifaceted challenge, with blood glucose control being a cornerstone in mitigating the risk of complications and enhancing the quality of life for individuals with diabetes [1]. Continuous glucose monitoring (CGM) has emerged as a transformative technology, offering real-time insights into glucose dynamics and enabling personalized adjustments to lifestyle and treatment regimens [2]. By providing continuous, real-time glucose data, CGM empowers individuals with diabetes to make informed decisions regarding their diet, exercise, and medication, ultimately aiming to optimize blood sugar control [3]. While evidence indicates a statistically significant reduction in glycated hemoglobin (HbA1c) levels, with an approximate decrease of 0.32%, associated with the use of CGM [4], several research gaps persist, necessitating further investigation. These gaps highlight the need to explore how CGM, when combined with educational or behavioral interventions, affects overall diabetes management. Of particular importance is understanding the physiological and behavioral impact of CGM. This involves evaluating factors such as glycemic control variables, patient medication adherence, the ability to integrate CGM into daily diabetes management routines, and the extent to which it promotes health behavior engagement in self-management efforts.

Self-management of diabetes requires theory-based patient education that drives patients toward effective disease management and improved health outcomes [5]. Self-regulation theory, rooted in social cognitive principles, outlines a patient-centered continuum of actions. It involves three fundamental processes: self-monitoring, self-judgment, and self-reaction. Through these processes, patients attain self-regulation, learn from their experiences, and continually adapt to achieve desired outcomes [6]. Several studies have applied self-regulation theory to diabetes management, demonstrating its efficacy in improving patient outcomes by emphasizing patient-driven problem-solving, fostering self-management, and facilitating sustainable health promotion beyond medical team interventions [7]. Disease control skills emerge through the process of self-regulation, wherein specific problematic behaviors are identified through self-monitoring, judgments are made based on observations, and appropriate responses are taken to meet desired goals, involving a continuous and reciprocal process [6]. Following a series of observations and judgments, self-reaction entails attempting to execute a specific plan to achieve desired outcomes. While CGM has been widely studied for its ability to improve glycemic outcomes, research on its influence on patient behaviors and adherence remains limited. Additionally, a meta-analysis found that only 6% of studies examined behavior change, such as self-efficacy, health behavior, and medication adherence [7], highlighting a significant gap in the literature. Despite the growing body of evidence supporting self-regulation mechanisms in health behavior change, there remains a notable gap in research specifically examining the application of self-regulation theory in the context of continuous glucose monitoring (CGM). This randomized controlled study aims to bridge this gap by investigating the application of self-regulation theory integrated with CGM to assess its effects on glycemic indices, self-efficacy, health-promoting behaviors, and medication adherence in patients with diabetes. 

Nurses play a critical role in diabetes management, serving as educators, advocates, and coordinators of care. By leveraging their unique position, nurses can implement evidence-based interventions such as self-regulation education and integrate technologies like CGM into patient care. This dual approach empowers patients to take control of their diabetes while enhancing self-management skills. Given the rising prevalence of diabetes, exploring how these combined strategies can support nursing practice and improve patient outcomes is crucial.

## 2. Materials and Methods

### 2.1. Design

This research employed a randomized, single-blind experimental design, where researchers assessing the outcomes were blinded, with pretest and post-test data collection methodologies structured as a parallel trial with a 1:1 allocation ratio.

### 2.2. Study Setting and Sampling

A randomized sample of diabetes patients who voluntarily enrolled from February 2023 to February 2024 was recruited from hospitals in northern Taiwan. Sample size calculation was determined based on the results of glycated hemoglobin (HbA1c) for independent t-tests, with an estimated effect size of d = 0.52 drawn from previous literature [8]. Using α = 0.05 and Power (1 − β) = 0.8, a sample size of 59 individuals (30 per group) was required. Considering a 10–15% dropout rate, the adjusted sample size ranges from 66 to 70 participants (33–35 per group). The calculation was performed using G*Power software (version 3.1.9.7, Heinrich Heine University, Düsseldorf, Germany) for Windows. The inclusion criteria for recruitment are as follows: (1) individuals diagnosed with Type 2 Diabetes Mellitus (T2DM) by a physician and aged 20 years or older; (2) HbA1c levels exceeding 8%, indicating suboptimal glycemic control; (3) use of a smartphone; and (4) comprehensive understanding and consent from participants regarding this study’s purpose and procedures. The exclusion criteria include (1) individuals with mental or cognitive impairments impeding communication and (2) diabetes health literacy scores below 9 points (comprising two parts, a cloze test and a reading comprehension test, totaling 12 questions and 12 points).

### 2.3. Randomization

This study adopted a single-blind research design, wherein neither the principal investigator nor co-investigator was engaged in the recruitment process. Participants were randomized using a computer-generated random allocation sequence created by an independent statistician not involved in the study. The randomization sequence was implemented using sealed, opaque, and consecutively numbered envelopes prepared before the start of the study. The envelopes were opened sequentially by a designated research assistant upon participant enrollment, who assigned them to either the CGM group or the control group. The principal investigators and assessors were blinded to the group assignments.

### 2.4. Interventions

The experimental group was fitted with continuous glucose monitoring (CGM) devices (Medtronic Guardian Connect System, approved under license number 032460 by the Taiwan Ministry of Health and Welfare) and received approximately one hour of guidance on self-regulation education. As the device would accompany the patients for seven days, daily phone calls were made to closely consult participants on self-regulation, including self-monitoring, self-judgment, and self-reaction, ensuring they could complete activities and help as needed. The control group received diabetes health education, along with verbal instructions on self-blood glucose monitoring and routine care.

The CGM group underwent an education program grounded in self-regulation, incorporating structured processes of self-monitoring, self-judgment, and self-reaction to promote self-regulatory behaviors for optimal glycemic control. Self-monitoring involved systematic tracking of diet, exercise, medication adherence, and lifestyle factors through daily activity logs, enabling participants to identify patterns and recognize glycemic fluctuations. Participants reflected on past challenges in diabetes management and used structured documentation to enhance self-awareness. Self-judgment focused on evaluating behaviors against clinical targets and personal health goals. Key components included comparing glucose levels with recommended standards, identifying behavioral patterns affecting glycemic control, and assessing the effectiveness of current habits. To facilitate this process, participants engaged in daily structured discussions (10–15 min) with the research team, reviewing their glucose data, identifying deviations, and refining self-assessment skills. Educational content on the glycemic index, portion control, and behavior analysis supported informed decision-making. In terms of self-reaction, participants were encouraged to explore appropriate responses to their judgment, such as adjusting portion sizes, selecting healthier food alternatives, engaging in timely physical activity, or modifying medication adherence as needed. To reinforce learning, participants received individualized feedback and goal-setting strategies to motivate and sustain behavior change.

Through ongoing 7-day monitoring of daily activities and real-time blood glucose responses, participants engaged in self-assessment to discern disparities between their actions and predetermined goals. This facilitated their self-regulation of health promotion strategies and behaviors, as illustrated in Figure 1. This approach underscores the significance of assessing incongruities between actions and objectives, ultimately leading to the refinement of health promotion strategies and behaviors while enhancing participants’ ability to make informed, personalized adjustments to their diabetes management.

There were no changes to the trial protocol after the commencement of this study. This trial adhered to the protocol approved by the Institutional Review Board, with all planned interventions and assessments executed as described.

### 2.5. Outcomes

In addition to demographics and diabetes-related characteristics, participants completed three scales: Medication Adherence, Diabetes Self-Efficacy, and Diabetes Health Behaviors. The demographics and diabetes-related characteristics included age, gender, education level, marital status, body mass index (BMI), duration of illness, diabetes complications, and blood glucose data (fasting blood glucose and glycated hemoglobin levels).

The 8-item Chinese Medication Adherence Scale developed by Ho and Lee [9] mainly assesses the extent of patient adherence to hypertension medication regimens encompassing factors such as frequency, type, and dosage of medication intake. Scores are derived from the frequency of medication-related behaviors observed within a week, with scores ranging from 0 (indicating non-compliance) to 5 (indicating full compliance), thereby reflecting varying levels of medication adherence. Regarding reliability and validity, the scale has been previously validated in the study by Ho and Lee [9], demonstrating strong psychometric properties. To adapt the scale for diabetes medication adherence, the wording of items was modified to expressly reflect diabetes-related medication behaviors while maintaining the original structure and measurement intent. In this study, Cronbach’s alpha exceeded 0.9 across all three-time points, indicating high internal consistency and reliability.

The Diabetes Self-Efficacy and Diabetes Health Behavior scales, developed by Chan et al. [10], aim to evaluate patients’ confidence levels and engagement in health behaviors pertaining to diabetes management. This instrument consists of 17 identical items, each assessed in two ways: (1) the Self-Efficacy Scale measures patients’ confidence in managing diabetes-related tasks. Participants rate how confident they feel about performing specific diabetes management behaviors on a 5-point scale (5 = Extremely confident, 1 = Not at all confident); (2) the Health Behaviors Scale assesses the frequency with which participants engage in the same diabetes-related behaviors. Responses are recorded on a 5-point scale (5 = Always, 1 = Never). Each scale generates independent scores. This dual assessment approach enables a comprehensive understanding of both perceived confidence and actual engagement in diabetes management behaviors. Higher scores on both scales indicate greater self-efficacy and more frequent engagement in health behaviors conducive to diabetes control. The questionnaire demonstrates expert validity (CVI = 0.92) and internal consistency reliability (Cronbach α = 0.92) [10]. In this study, Cronbach α for both scales were over 0.8 across three-time points.

### 2.6. Data Collection

Three specific time points for data collection to assess the impact of the intervention were conducted, including baseline (T0, pre-intervention), midpoint assessment at 8 days (T1, post-intervention completion), and follow-up at three months (T2, maintenance phase). To minimize assessor bias and ensure standardized data collection, an online survey was utilized.

### 2.7. Ethical Considerations

The formal commencement of this study occurred only after obtaining approval from the Institutional Review Board of Chang Gung Medical Foundation. Prior to this study’s initiation, prospective participants received comprehensive explanations regarding this study’s purpose, procedures, content, and anticipated time commitment. Their rights and interests were discussed thoroughly, and informed consent was obtained through signature before participation. Research data were anonymized using numerical coding, strictly protected, and utilized solely for academic research purposes, with no public disclosure. Participants retained the right to withdraw from this study at any point without consequence to their future medical care or other rights. Furthermore, in consideration of ethical principles, the rights of individuals in the control group, often overlooked in research processes, were respected. Upon unblinding, control group members were afforded the same benefits as those in the CGM group, ensuring equity in treatment.

### 2.8. Data Analysis

Quantitative data were analyzed using IBM^®^ SPSS^®^ version 28.0 for Windows. A per-protocol (PP) analysis was chosen to ensure that the results reflected the effects of the intervention on participants who adhered to the study protocol. The analysis focused on the distributions of both categorical and continuous variables, encompassing frequencies, means, standard deviations, and percentages. Baseline comparisons, including demographic characteristics and outcome measures, were conducted using the independent t-test and chi-square test. To evaluate the effects of the intervention within groups, paired samples t-tests were utilized. The generalized estimating equation (GEE) method was employed to assess the impact of the intervention between groups. Statistical significance for the outcome variables was set at *p* ≤ 0.05. G*Power for Windows (version 3.1.9.7) was used to calculate effect size and statistical power, with effect sizes interpreted as small (dz = 0.2), medium (dz = 0.5), and large (dz = 0.8), and a minimum acceptable statistical power level of 0.80 [11].

## 3. Results

### 3.1. Demographics of Sample and Baseline Comparison

In total, 120 participants underwent eligibility assessment, with 52 subsequently excluded from this study. Among these exclusions, 18 participants declined to participate, 16 were excluded due to changes in prescribed medication, 10 did not use smartphones, and 6 failed to meet the health literacy criteria as per the inclusion test. Following a 3-month follow-up period, two participants were lost to follow-up, attributed to non-attendance at the diabetes clinics (Figure 2). Of the 66 participants, 41 were male (62.12%) and 25 were female (37.88%). Their ages ranged from 29 to 80 years old (*M* = 55.82, *SD* = 0.97).

Between-group comparisons of baseline data revealed no significant differences in demographic and diabetes-related characteristics, including age, gender, education, marital status, duration of diabetes, and presence of diabetes complications. Additionally, the mean summed scores on measures of medication adherence, diabetes self-efficacy, diabetes health behaviors, HbA1c levels, fasting blood glucose, and BMI demonstrated homogeneity between the groups (Table 1).

### 3.2. Comparisons of Outcomes Between and Within Groups

At the different time points, no significant differences were observed between the CGM and control groups regarding medication adherence, diabetes self-efficacy, diabetes health behaviors, HbA1c (%), and fasting glucose (mg/dL). Within-group analyses further revealed no significant differences in medication adherence scale or fasting blood glucose levels across time points (T0, T1, and T2) for either group.

However, a notable difference was found in diabetes self-efficacy and health behaviors. The control group exhibited a significant decline in self-efficacy scores, dropping from 72.70 to 68.21, representing a 6.18% decrease (*p* = 0.018). Conversely, the CGM group experienced a slight but non-significant reduction in self-efficacy, with scores declining from 71.82 to 70.61 (a 1.68% decrease, *p* = 0.647). In terms of diabetes health behaviors, the control group displayed a slight but statistically non-significant increase, with scores rising from 60.52 to 63.79 (a 5.40% increase, *p* = 0.061). In contrast, the CGM group demonstrated a significant improvement in health behaviors, with scores increasing from 61.61 to 66.42, marking a 7.81% increase (*p* = 0.010) and an effect size of 0.47, indicating a moderate practical impact.

A similar trend was observed in HbA1c levels. The control group displayed a slight but statistically non-significant reduction in HbA1c from 9.45 to 9.08 (a 3.92% decrease, *p* = 0.289). Conversely, the CGM group demonstrated a significant reduction in HbA1c levels, declining from 9.37 to 8.79 (a decrease of 6.19%, *p* = 0.017) and reaching an effect size of 0.83. (Table 2 and Figure 3).

### 3.3. Harms or Unintended Effects

No major safety concerns or adverse effects were reported during the trial. However, two participants experienced issues related to CGM devices, including malfunctions or adhesive failures, which necessitated a reinsertion of the device. These minor issues were addressed promptly and did not lead to withdrawal from the study.

## 4. Discussion

This preliminary study evaluates the effectiveness of integrating self-regulation education with continuous glucose monitoring (CGM) in diabetes management. Unlike previous studies [12,13], which primarily focused on clinical outcomes like HbA1c and glucose variation, this study also examines behavioral dimensions, including diabetes self-efficacy and health behaviors. The findings highlight that while no immediate differences were observed between the CGM and control groups, CGM use contributed to significant improvements in diabetes health behaviors, with an effect size approaching a moderate level, and glycemic control, with a large effect size over time. These results underscore the potential of CGM as a valuable tool not only for enhancing clinical outcomes but also for fostering positive behavioral changes in diabetes management.

While the CGM group consistently maintained their diabetes self-efficacy scores throughout this study, a notable decline was observed in the control group at the third time point (T2). Interestingly, these findings contrast with those reported in a systematic review [14] which highlighted a significant and positive impact of diabetes education self-management interventions on enhancing self-efficacy among patients with type 2 diabetes. The challenge of sustaining or enhancing self-efficacy following diabetes education interventions can be multifaceted. One possible explanation for the decline in the control group is an initial overestimation of self-efficacy, a phenomenon observed in previous research on diabetes self-management programs [15]. As participants engaged in the diabetes education program and closely monitored their behaviors over time, they gained a more realistic understanding of their actual capabilities and encountered the inherent challenges of diabetes management. This increased awareness may have contributed to the observed decline in self-efficacy scores among the control group. Additionally, the observed difference between groups suggests that the CGM intervention may have played a role in sustaining self-efficacy levels. Unlike the control group, participants in the CGM group received continuous real-time feedback on glucose fluctuations, allowing them to make immediate behavioral adjustments and observe the direct impact of their actions. This reinforcement mechanism may have enhanced their confidence in managing their condition, helping them sustain their self-efficacy scores despite the complexities of diabetes self-management. In contrast, the control group lacked this immediate feedback loop, which may have contributed to a gradual decline in their perceived ability to manage diabetes effectively.

The discrepancy in health behaviors observed in both groups highlights the varying impact of interventions on diabetes management. The CGM group demonstrated a significant improvement in diabetes health behaviors after three months, whereas the control group exhibited no significant differences across time points. This finding indicates that the integration of CGM with self-regulation education may facilitate the adoption of health-promoting behaviors, potentially reinforcing the application of diabetes management strategies in daily life. The ability to receive real-time glucose feedback may have played a crucial role in fostering behavioral adjustments and supporting long-term adherence to health-promoting practices. These results align with evidence from a review study [16], further emphasizing the value of incorporating CGM as a complementary tool in diabetes self-management interventions. Moreover, the significant increase in health behavior scores observed in the CGM group suggests that participants not only acquired relevant knowledge and skills through the education program but also actively translated them into improved diabetes management practices, highlighting the potential of CGM to bridge the gap between education and sustained behavior change.

A parallel trend was observed between improvements in diabetes health behaviors and reductions in HbA1c levels. The CGM group consistently demonstrated significant enhancements in both diabetes health behaviors and HbA1c levels over time, whereas the control group showed no significant changes. The significance observed in this study between improved diabetes health behaviors and better glycemic control suggests that enhancements in health behaviors may contribute to lower HbA1c levels. Real-time CGM provides frequent blood glucose monitoring, enabling timely adjustments in diet, exercise, and medication, which contributes to better long-term glycemic control and reduced HbA1c levels [12]. This underscores the potential clinical benefits of CGM alongside self-regulation education in achieving glycemic control outcomes among patients with diabetes.

In this study, within-group analyses indicated consistent medication adherence over time for both groups. A potential reason for this could be the ceiling data resulting from high baseline adherence. Participants began this study with medication adherence scores that were already near the maximum possible, leaving minimal room for improvement. This high initial adherence level makes it challenging to detect any significant improvements or differences between the intervention and control groups [17]. Even if the intervention had a positive effect, the high baseline scores likely obscured observable changes within the scoring range of the scale used [18].

Our study illustrates the complex and dynamic interplay between behavioral and clinical outcomes in diabetes management. The combination of CGM and self-regulation education demonstrates potential benefits in maintaining diabetes self-efficacy, promoting health behaviors, and improving glycemic control. However, no significance between groups may be attributed to several factors, including the short intervention duration, potential variability in participant adherence, psychological impact, and the limited timeframe for behavioral changes to produce distinct glycemic effects. To improve future studies, integrating more sensitive physiological, psychological, and behavioral measures such as continuous glucose monitoring (CGM) data over an extended period, may provide a more precise understanding of intervention effects across time points and groups.

Nurses are pivotal in delivering patient-centered care, providing education, and supporting behavioral changes essential for effective diabetes management. The improvement in health behaviors and glycemic control observed in the CGM group reflects the potential of nurse-led interventions to promote patient engagement and adherence to self-management plans. Moreover, the sustained self-efficacy observed in the CGM group highlights the need for nursing strategies that emphasize ongoing support and reinforcement. Finally, developing robust educational programs that focus on self-regulation and the use of CGM can empower patients to manage their diabetes better. Policies supporting nurse-led educational programs and follow-up systems could further enhance patient outcomes and optimize diabetes management practices.

### Limitations and Recommendations for Further Research

One limitation of this study was selection bias, which limited the external validity of this study and suggested the generalizability of the findings to broader populations with diabetes. This study’s findings were most relevant to a specific subset of the diabetes population: those with higher HbA1c levels, higher health literacy, and access to technology. In addition, this study included only participants who completed the study according to the protocol, potentially overestimating the intervention effect. However, given the small number of dropouts (one per group), the impact on statistical power was minimal. Future studies should consider with more diverse, larger sample sizes and higher dropout rates should consider Intention-to-Treat (ITT) analysis with appropriate imputation techniques to strengthen the generalizability of findings. Additionally, the short intervention period (7 days) and limited CGM duration (7 days) may have introduced recall bias as participants relied on memory. This limited timeframe could influence the overall impact and sustainability of the self-regulation education combined with the CGM intervention. Future studies should consider extending the intervention period and CGM use to evaluate the sustainability of self-regulation behaviors over time. Furthermore, as an outpatient study, despite being a randomized controlled trial (RCT), the results might be influenced by factors such as variability in participants’ adherence to the intervention, differing levels of social support at home, and potential discrepancies in psychological status. Further research should consider these factors and potentially include more extended follow-up periods and additional support mechanisms to better understand the intervention’s effectiveness in a real-world outpatient setting. Overall, future studies should address several limitations observed in this study, including broader inclusion criteria, extended duration of CGM use, and consideration of various settings.

## 5. Conclusions

This study underscores the potential benefits of combining continuous glucose monitoring (CGM) with self-regulation education in diabetes management. While no differences in medication adherence, diabetes self-efficacy, diabetes health behaviors, and glycemic control after the intervention were observed between the CGM and control groups, the CGM group demonstrated sustained engagement in diabetes management and significant improvements in both diabetes health behaviors and HbA1c levels over time. These findings highlight the importance of personalized, patient-centered approaches, highlighting the need for ongoing support and reinforcement through self-regulation education. Additionally, the role of the CGM device in providing real-time feedback appears to be a crucial factor in facilitating behavioral adjustments and improving long-term outcomes. Overall, this study provides valuable insights into the advantages of integrating CGM with self-regulation education, advocating for continuous education and individualized care strategies to optimize diabetes self-management and achieve better glycemic control.

## Figures and Tables

**Figure 1 nursrep-15-00094-f001:**
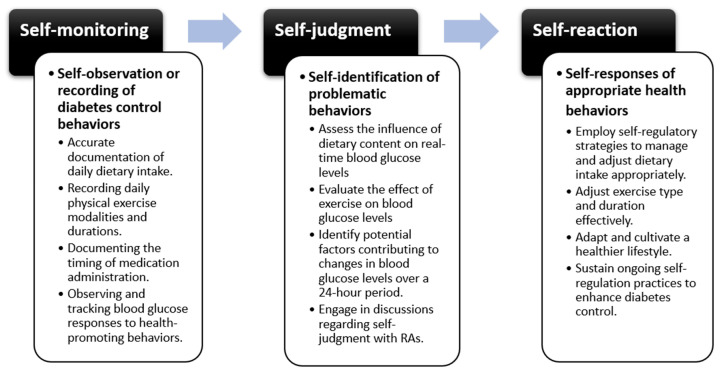
The process of self-regulation education in diabetes management.

**Figure 2 nursrep-15-00094-f002:**
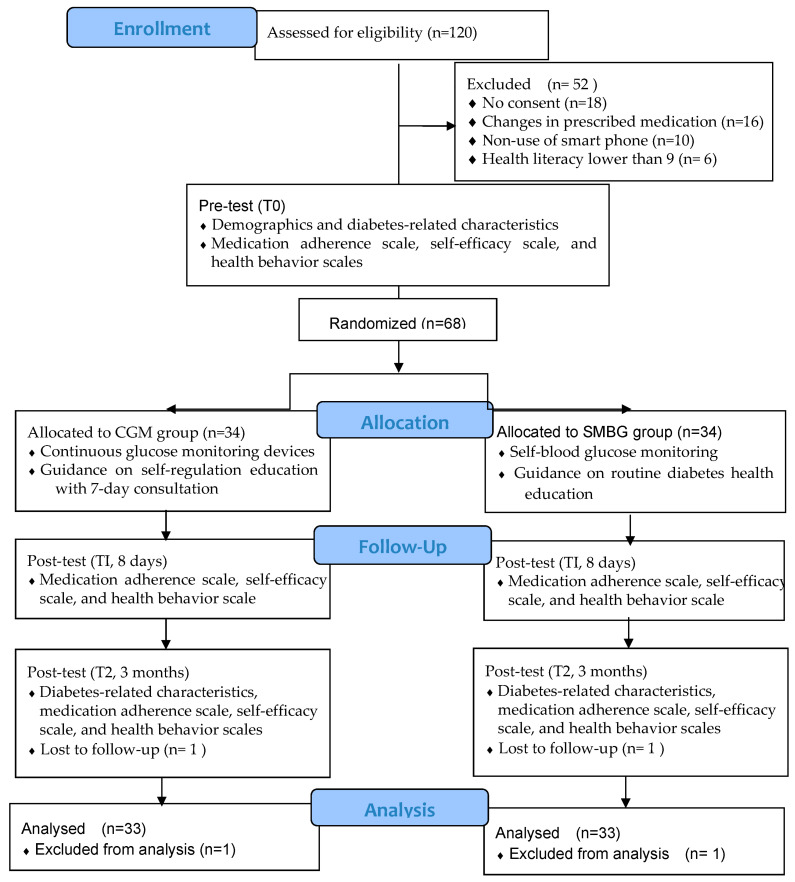
Flow diagram of data collection.

**Figure 3 nursrep-15-00094-f003:**
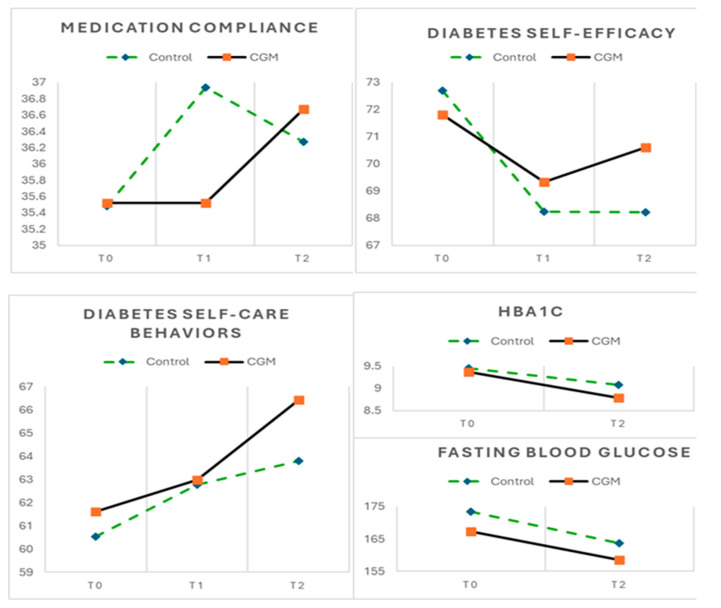
Mean scores of outcomes across 3 time periods for CGM (n = 33) and control groups (n = 33).

**Table 1 nursrep-15-00094-t001:** Demographic characteristics.

Variables	Total	Control		CGM		*x* ^2^	*p*
		N	%	N	%		
Gender						0.6	0.800
Male	41 (62.12)	20	60.6	21	63.6		
Female	25 (37.88)	13	39.4	12	36.4		
Education						0.6	0.800
≤High school	41 (62.12)	21	63.6	20	60.6		
>High school	25 (37.88)	12	36.4	13	39.4		
Marriage status						0.56	0.453
Married	39 (59.09)	21	63.6	18	54.5		
Single or others	27 (40.91)	12	36.4	15	45.5		
Diabetes Complications						3.27	0.071
No	43 (65.15)	18	54.5	25	75.8		
Yes	23 (34.85)	15	45.5	8	24.2		
		*M*	*SD*	*M*	*SD*	*t*	*p*
Age		55.82	11.21	51.76	12.74	1.37	0.174
Duration of diabetes		9.64	6.75	9.56	6.81	0.05	0.964
BMI		27.96	4.79	27.10	4.80	0.67	0.506
Medication adherence		35.48	6.14	35.52	4.94	−0.02	0.982
Diabetes self-efficacy		72.70	13.00	71.82	11.75	−0.39	0.699
Diabetes health behaviors		60.52	11.19	61.61	10.47	−0.41	0.684
HbA1c (%)		9.45	1.34	9.37	1.45	0.23	0.820
Fasting glucose (mg/dl)		173.50	61.26	167.33	66.27	0.39	0.698

*M* = mean; *SD* = standard deviation.

**Table 2 nursrep-15-00094-t002:** Comparisons of outcomes between groups and within groups.

	T0		T1		T0-T1	Time × Group	T2		T0-T2	Time × Group
	*M*	*SD*	*M*	*SD*	*t* ^1^/*p*	B ^2^/*p*	*M*	*SD*	*t* ^1^/*p*	B ^2^/*p*
Medication adherence						−1.52/0.466				0.36/0.758
Control (n = 33)	35.48	6.14	36.94	11.25	−0.84/0.406		36.27	5.50	−0.87/0.389	
CGM (n = 33)	35.52	4.94	35.52	5.77	0.00/1.000		36.67	4.65	−1.46/0.153	
Diabetes self-efficacy						−1.06/0.656				1.55/0.519
Control	72.70	13.00	68.25	11.45	1.82/0.078		68.21	13.65	2.50/0.018	
CGM	71.82	11.75	69.33	11.07	1.10/0.279		70.61	10.38	0.46/0.647	
Diabetes health behaviors										
Control	60.52	11.19	62.78	10.25	−1.53/0.136	1.66/0.597	63.79	10.48	−1.94/0.061	3.28/0.295
CGM	61.61	10.47	62.97	10.55	−0.79/0.433		66.42	10.35	−2.74/0.010	
HbA1c						Nil				−0.21/0.552
Control	9.45	1.34	Nil	Nil	Nil		9.08	1.73	1.08/0.289	
CGM	9.37	1.45	Nil	Nil	Nil		8.79	1.76	2.51/0.017	
Fasting blood glucose						Nil				0.89/0.958
Control	173.50	61.26	Nil	Nil	Nil		163.72	45.40	0.54/0.593	
CGM	167.33	66.27	Nil	Nil	Nil		158.44	61.12	0.68/0.503	

^1^ Paired *t*-test; ^2^ generalized estimating equation (GEE) statistics.

## Data Availability

Data is contained within the article.

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
