# Peer review of "The Impact of Self-Regulation Education Combined with Continuous Glucose Monitoring (CGM) on Diabetes Outcomes: A Randomized Controlled Study"

_nursrep, 2025, doi:10.3390/nursrep15030094_

Round 1
Reviewer 1 Report
Comments and Suggestions for Authors
Thank you for the opportunity to review this paper. The results of this study reveal the benefits of CGM, which can be applied to patients who struggle to control their diabetes. Additionally, I have outlined comments for the authors' consideration below:
-
Could you provide additional information about the participants included in the study, specifying whether they are patients with T2DM, T1DM, or both? This clarification is important, as most individuals diagnosed with diabetes before the age of 20 are typically T1DM patients.
-
This study includes participants aged 20 years and older. While the selection criteria consider factors such as technology use or health literacy, age differences may influence participants' ability to regulate their blood glucose levels, potentially affecting the study's outcomes.
-
Could you provide more details on self-reaction processes, which play a crucial role in helping individuals lower their blood glucose levels after becoming aware of their blood glucose status?
-
Regarding "The Diabetes Self-Efficacy and Diabetes Health Behavior Scales," developed by Chan et al., the instrument comprises 17 items divided into two components: the self-efficacy scale and the health behaviors scale, each rated on a 1–5 scale. Therefore, the total score ranges from 17 to 85. However, the study results in Table 1 report separate mean scores for the components: diabetes self-efficacy (71.82–72.70) and diabetes health behaviors (60.52–61.61). Could you clarify how many items are included in each component?
Author Response
Dear reviewer 1
We sincerely appreciate the time and effort that you and the reviewers have invested in evaluating our manuscript, titled “ The Impact of Self-Regulation Education Combined with Continuous Glucose Monitoring (CGM) on Diabetes Outcomes: A Randomized Controlled Study.” We are grateful for the constructive feedback and insightful comments, which have helped us improve the quality and clarity of our work. We have carefully addressed each comment, and the corresponding revisions are incorporated into the revised manuscript.
Below, we provide a point-by-point response to each reviewer’s comment. The reviewers' comments are in bold black, and our responses are provided below each in red.
Comments 1: Could you provide additional information about the participants included in the study, specifying whether they are patients with T2DM, T1DM, or both? This clarification is important, as most individuals diagnosed with diabetes before the age of 20 are typically T1DM patients.
Response 1: Agree.
We have revised the manuscript to clarify that the study participants were exclusively individuals diagnosed with Type 2 Diabetes Mellitus (T2DM). Specifically, in the Methods section under Study Setting and Sampling, we modified the inclusion criteria to state that only patients with T2DM were included explicitly. The updated text can be found on page [3], paragraph [2], lines [97]:
"The inclusion criteria for recruitment are as follows: (1) individuals diagnosed with Type 2 Diabetes Mellitus (T2DM) by a physician and aged 20 years or older..."
Comments 2: This study includes participants aged 20 years and older. While the selection criteria consider factors such as technology use or health literacy, age differences may influence participants' ability to regulate their blood glucose levels, potentially affecting the study's outcomes.
Response 2: Agree.
While this study included participants aged 20 years and older, potential variations in age-related factors, such as cognitive function, technological adaptability, and physiological glucose regulation, may have influenced individual responses to the intervention. However, to minimize the impact of these differences, we controlled for diabetes health literacy by including only participants with a “literacy score of 9 or above” in the exclusive criteria on page [3], paragraph [2], lines [97-104]. Furthermore, “baseline characteristics”, including age and HbA1c levels, were statistically comparable between groups, ensuring that age-related disparities did not confound any observed intervention effects in the result on page [3], paragraph [2], lines [97-104].
Comments 3: Could you provide more details on self-reaction processes, which play a crucial role in helping individuals lower their blood glucose levels after becoming aware of their blood glucose status?
Response 3: Agree.
We have further revised the Interventions section on page [3], paragraph [4], lines [120-138], where we expanded the explanation of self-reaction as part of self-regulation education. The updated text reads:
“The CGM group underwent an education program grounded in self-regulation, incorporating structured processes of self-monitoring, self-judgment, and self-reaction to promote self-regulatory behaviors for optimal glycemic control. Self-monitoring involved systematic tracking of diet, exercise, medication adherence, and lifestyle factors through daily activity logs, enabling participants to identify patterns and recognize glycemic fluctuations. Participants reflected on past challenges in diabetes management and used structured documentation to enhance self-awareness. Self-judgment focused on evaluating behaviors against clinical targets and personal health goals. Key components included comparing glucose levels with recommended standards, identifying behavioral patterns affecting glycemic control, and assessing the effectiveness of current habits. To facilitate this process, participants engaged in daily structured discussions (10–15 minutes) with the research team, reviewing their glucose data, identifying deviations, and refining self-assessment skills. The educational content on the glycemic index, portion control, and behavior analysis supported informed decision-making. In terms of self-reaction, participants were encouraged to explore appropriate responses to their judgment, such as adjusting portion sizes, selecting healthier food alternatives, engaging in timely physical activity, or modifying medication adherence as needed. To reinforce learning, participants received individualized feedback and goal-setting strategies to motivate and sustain behavior change.”
Comments 4: Regarding "The Diabetes Self-Efficacy and Diabetes Health Behavior Scales," developed by Chan et al., the instrument comprises 17 items divided into two components: the self-efficacy scale and the health behaviors scale, each rated on a 1–5 scale. Therefore, the total score ranges from 17 to 85. However, the study results in Table 1 report separate mean scores for the components diabetes self-efficacy (71.82–72.70) and diabetes health behaviors (60.52–61.61). Could you clarify how many items are included in each component?
Response 4: Agree.
We have addressed the measurement structure on page [4], paragraph [4], lines [162-189]:
“This instrument consists of 17 identical items, each assessed in two ways: (1) Self-Efficacy Scale – Measures patients' confidence in managing diabetes-related tasks. Participants rate how confident they feel about performing specific diabetes management behaviors on a 5-point scale (5 = Extremely confident, 1 = Not at all confident); (2) Health Behaviors Scale – Assesses the frequency with which participants engage in the same diabetes-related behaviors. Responses are recorded on a 5-point scale (5 = Always, 1 = Never). Each scale generates independent scores. This dual-assessment approach enables a comprehensive understanding of both perceived confidence and actual engagement in diabetes management behaviors.”
Dear reviewer 1
We sincerely appreciate the time and effort that you and the reviewers have invested in evaluating our manuscript, titled “ The Impact of Self-Regulation Education Combined with Continuous Glucose Monitoring (CGM) on Diabetes Outcomes: A Randomized Controlled Study.” We are grateful for the constructive feedback and insightful comments, which have helped us improve the quality and clarity of our work. We have carefully addressed each comment, and the corresponding revisions are incorporated into the revised manuscript.
Below, we provide a point-by-point response to each reviewer’s comment. The reviewers' comments are in bold black, and our responses are provided below each in red.

Reviewer 2 Report
Comments and Suggestions for Authors
This study seems to be worthwhile in that it developed a program using CGM for diabetic patients. However, the results are somewhat poor, and the reason for this seems to be that the process was only applied for a short period of 7 days. In order to praise the researchers' efforts and make it possible for publication, it seems necessary to organize and rewrite the somewhat confusing and inaccurate content neatly.
- There must be other studies related to this theory. Please introduce some of them.
- What are the research variables that are mainly examined for change in experimental studies related to CGM?
- Why do you think these variables are important? We need a description of the basis for why these variables were used in this study.
- Why did you set the HbA1C standard at 8% or higher when selecting the subjects?
- Does your research also contained the participants with type 1 Diabetes? If not, you should write down this standard.
- Please introduce what elements are included in self-judgement and what elements are included in self-reaction. In other words, a more detailed explanation is needed on what basis the contents of the intervention were developed and how the educational content was selected and written.
- Please write down when each of the three time points used in the study was and how the data was collected
5.Because of your short time intervention period, the results of the dependent variable may have been biased due to memory. Please write down this point in the limitations and provide directions for what to do in future studies.
- In line 280, it says correlation. When was this analyzed? Was it analyzed in this study?
- Write why the interaction between group and time did not appear and what needs to be improved in future studies.
- In lines 235-254, it is difficult to accept why this phenomenon only appeared in the control group. Are there any other reasons? Other reasons such as the experimental intervention being more powerful?
Author Response
Dear reviewer 2
We sincerely appreciate the time and effort that you and the reviewers have invested in evaluating our manuscript, titled “ The Impact of Self-Regulation Education Combined with Continuous Glucose Monitoring (CGM) on Diabetes Outcomes: A Randomized Controlled Study.” We are grateful for the constructive feedback and insightful comments, which have helped us improve the quality and clarity of our work. We have carefully addressed each comment, and the corresponding revisions are incorporated into the revised manuscript.
Below, we provide a point-by-point response to each reviewer’s comment. The reviewers' comments are in bold black, and our responses are provided below each in red.
Introduction>
Comments 1: There must be other studies related to this theory. Please introduce some of them.
Response 1: Thank you for your insightful comment. We appreciate the opportunity to clarify this point. As noted, several studies have applied self-regulation theory to diabetes management, demonstrating its efficacy in improving patient outcomes. The systematic meta-review by Hennessy et al. (2020) has comprehensively reviewed studies related to self-regulation mechanisms in health behavior change, including applications in diabetes management. However, upon reviewing the existing literature, we found that no studies have specifically examined the application of self-regulation theory in the context of continuous glucose monitoring (CGM). To address this gap, our study aims to explore how integrating self-regulation theory with CGM technology may influence diabetes self-management, medication adherence, and glycemic control. We have revised the manuscript accordingly to enhance clarity. The updated text can be found on page [2], paragraph [2], lines [57-60 & 66-73]:
“Several studies have applied self-regulation theory to diabetes management, demonstrating its efficacy in improving patient outcomes by emphasizing patient-driven problem-solving, fostering self-management, and facilitating sustainable health promotion beyond medical team interventions [7]” & “Additionally, despite the growing body of evidence supporting self-regulation mechanisms in health behavior change, there remains a notable gap in research explicitly examining the application of self-regulation theory in the context of continuous glucose monitoring (CGM). “
Comments 2: What are the research variables that are mainly examined for change in experimental studies related to CGM?
Response 2: Thank you for your insightful question regarding the research variables primarily examined for change in experimental studies related to CGM. In response, we have clarified the key variables assessed in the introduction, focusing on the physiological and behavioral impact of CGM, including glycemic control and behavioral/adherence-related factors. The updated text can be found on page [2], paragraph [2], lines [57-60 & 66-73]:
“Of particular importance is understanding the physiological and behavioral impact of CGM. This involves evaluating factors such as glycemic control variables, patient medication adherence, the ability to integrate CGM into daily diabetes management routines, and the extent to which it promotes health behavior engagement in self-management efforts.
Comments 3: Why do you think these variables are important? We need a description of the basis for why these variables were used in this study.
Response 3: Thank you for your comment. We appreciate the opportunity to clarify the basis for selecting these variables in our study. As mentioned in the manuscript, a meta-analysis found that only 6% of studies examined behavior change variables such as self-efficacy, health behavior, and medication adherence. This gap highlights the need to investigate behavioral and adherence-related factors alongside glycemic control variables to understand CGM’s impact on diabetes management better. Therefore, we have expanded the content on page [2], paragraph [2], lines [57-64]
“While CGM has been widely studied for its ability to improve glycemic outcomes, research on its influence on patient behaviors and adherence remains limited. Additionally, a meta-analysis found that only 6% of studies examined behavior change, such as self-efficacy, health behavior, and medication adherence [7], highlighting a significant gap in the literature. Despite the growing body of evidence supporting self-regulation mechanisms in health behavior change, there remains a notable gap in research specifically examining the application of self-regulation theory in the context of continuous glucose monitoring (CGM).”
Comments 1: Why did you set the HbA1C standard at 8% or higher when selecting the subjects?
Response 1: Thank you for your insightful comment. We selected HbA1c ≥ 8% as the inclusion criterion to identify participants with poor glycemic control. As the American Diabetes Association (ADA) defines poor glycemic control as HbA1c levels ≥8%, this threshold is commonly used in clinical and research settings to indicate suboptimal diabetes management.
Comments 2: Does your research also contain the participants with type 1 Diabetes? If not, you should write down this standard.
Response 2: Agree.
We have revised the manuscript to clarify that the study participants were exclusively individuals diagnosed with Type 2 Diabetes Mellitus (T2DM). Specifically, in the Methods section under Study Setting and Sampling, we modified the inclusion criteria to state that only patients with T2DM were included explicitly. The updated text can be found on page [3], paragraph [2], lines [96-103]
"The inclusion criteria for recruitment are as follows: (1) individuals diagnosed with Type 2 Diabetes Mellitus (T2DM) by a physician and aged 20 years or older..."
Comments 3: Please introduce what elements are included in self-judgment and what elements are included in self-reaction. In other words, a more detailed explanation is needed on what basis the contents of the intervention were developed and how the educational content was selected and written.
Response 3: Thank you for your valuable feedback. We have revised the manuscript to explicitly define the key components of self-judgment and self-reaction within the self-regulation framework. Specifically, self-judgment now includes comparison against clinical standards, pattern recognition, and behavioral assessment, while self-reaction encompasses corrective actions, emotional regulation, and reinforcement of adaptive behaviors. The updated text can be found on page [3], paragraph [4], lines [120-138] “The CGM group underwent an education program grounded in self-regulation, incorporating structured processes of self-monitoring, self-judgment, and self-reaction to promote self-regulatory behaviors for optimal glycemic control. Self-monitoring involved systematic tracking of diet, exercise, medication adherence, and lifestyle factors through daily activity logs, enabling participants to identify patterns and recognize glycemic fluctuations. Participants reflected on past challenges in diabetes management and used structured documentation to enhance self-awareness. Self-judgment focused on evaluating behaviors against clinical targets and personal health goals. Key components included comparing glucose levels with recommended standards, identifying behavioral patterns affecting glycemic control, and assessing the effectiveness of current habits. To facilitate this process, participants engaged in daily structured discussions (10–15 minutes) with the research team, reviewing their glucose data, identifying deviations, and refining self-assessment skills. The educational content on the glycemic index, portion control, and behavior analysis supported informed decision-making. In terms of self-reaction, participants were encouraged to explore appropriate responses to their judgment, such as adjusting portion sizes, selecting healthier food alternatives, engaging in timely physical activity, or modifying medication adherence as needed. To reinforce learning, participants received individualized feedback and goal-setting strategies to motivate and sustain behavior change.”
Comments 4: Please write down when each of the three-time points used in the study was and how the data was collected
Response 4: Thank you for your valuable comment. We have added a subtitle of data collection on page [5], paragraph [2], lines [185-189]. “Three specific time points for data collection to assess the impact of the intervention were conducted, including baseline (T0, pre-intervention), midpoint assessment at 8 days (T1, post-intervention completion), and follow-up at three months (T2, maintenance phase). To minimize assessor bias and ensure standardized data collection, an online survey was utilized.”
Comments 5: Because of your short time intervention period, the results of the dependent variable may have been biased due to memory. Please write down this point in the limitations and provide directions for what to do in future studies.
Response 4: We have revised the limitation focus on recall bias on page [11], paragraph [3], lines [357-361] “Additionally, the short intervention period (7 days) and limited CGM duration (7 days) may have introduced recall bias as participants relied on memory. This limited timeframe could influence the overall impact and sustainability of the self-regulation education combined with the CGM intervention. Future studies should consider extending the intervention period and CGM uses to evaluate the sustainability of self-regulation behaviors over time.”
Comments 1: In line 280, it says correlation. When was this analyzed? Was it explored in this study?
Response 1: Thank you for your careful review. The use of the word "correlation" in line 280 was incorrect. We intended to indicate that the two results showed significant differences rather than implying a correlation analysis. We have revised the wording accordingly to reflect the statistical findings accurately. The updated text can be found on page [11], paragraph [2], lines [311-313]: “ The significance observed in this study between improved diabetes health behaviors and better glycemic control suggests that enhancements in health behaviors may contribute to lower HbA1c levels.”
Comments 2: Write why the interaction between group and time did not appear and what needs to be improved in future studies.
Response 2: Thank you for your valuable comment. We have added more discussion on page [11], paragraph [2], lines [330-337] “However, no significance between groups may be attributed to several factors, including the short intervention duration, potential variability in participant adherence, psychological impact, and the limited timeframe for behavioral changes to produce distinct glycemic effects. To improve future studies, integrating more sensitive physiological, psychological, and behavioral measures, such as continuous glucose monitoring (CGM) data over an extended period, may provide a more precise understanding of intervention effects across time points and groups.”
Comments 3: In lines 235-254, it is difficult to accept why this phenomenon only appeared in the control group. Are there any other reasons? Other reasons such as the experimental intervention being more powerful?
Response 3: Thank you for your valuable comment. We have revised the discussion on page [10], paragraph [2], lines [277-292] “One possible explanation for this decline is the initial overestimation of self-efficacy among participants in the control group, a phenomenon noted in prior research on diabetes self-management programs [16]. As participants engaged in the diabetes education program and vigilantly monitored their behaviors over time, they likely gained a more realistic understanding of their actual capabilities and encountered the inherent challenges of diabetes management. Consequently, this increased awareness may have contributed to the observed decline in self-efficacy scores among the control group. Additionally, the observed difference between groups suggests that the CGM intervention may have played a role in sustaining self-efficacy levels. Unlike the control group, participants in the CGM group received continuous real-time feedback on glucose fluctuations, allowing them to make immediate behavioral adjustments and observe the direct impact of their actions. This reinforcement mechanism may have enhanced their confidence in managing their condition, helping them sustain their self-efficacy scores despite the complexities of diabetes self-management. In contrast, the control group lacked this immediate feedback loop, which may have contributed to a gradual decline in their perceived ability to manage diabetes effectively.”

Round 2
Reviewer 2 Report
Comments and Suggestions for Authors
All recommendations were revised correctly. However, please see the table 1.
The absence or presence of 0 before the decimal point should match.
Thank you.
Comments on the Quality of English Language
The English can be improved.
Author Response
Dear reviewer 2
We sincerely appreciate the time and effort that you have invested in evaluating our manuscript. We are grateful for the constructive feedback and insightful comments, which have helped us improve the quality and clarity of our work. We have carefully addressed each comment, and the corresponding revisions are incorporated into the revised manuscript.
Below, we provide a point-by-point response to the reviewer’s comment. The reviewers' comments are in bold black, and our responses are provided below in red.
Comment 1: However, please see the table 1. The absence or presence of 0 before the decimal point should match.
Response: We have revised Table 1 and Table 2
Table 2 Comparisons of outcomes between groups and within groups
|
T0 |
T1 |
T0-T1 |
Group*effects |
T2 |
T0-T2 |
Group*effects |
|||
M |
SD |
M |
SD |
t1 /p |
B2/p |
M |
SD |
t1/p |
B2/p |
|
Medication adherence |
|
|
|
|
|
-1.52/0.466 |
|
|
|
0.36/0.758 |
Control (n=33) |
35.48 |
6.14 |
36.94 |
11.25 |
-0.84/0.406 |
|
36.27 |
5.50 |
-0.87/0.389 |
|
CGM (n=33) |
35.52 |
4.94 |
35.52 |
5.77 |
0.00/1.000 |
|
36.67 |
4.65 |
-1.46/0.153 |
|
Diabetes self-efficacy |
|
|
|
|
|
-1.06/0.656 |
|
|
|
1.55/0.519 |
Control |
72.70 |
13.00 |
68.25 |
11.45 |
1.82/0.078 |
|
68.21 |
13.65 |
2.50/0.018 |
|
CGM |
71.82 |
11.75 |
69.33 |
11.07 |
1.10/0.279 |
|
70.61 |
10.38 |
0.46/0.647 |
|
Diabetes health behaviors |
|
|
|
|
|
|
|
|
|
|
Control |
60.52 |
11.19 |
62.78 |
10.25 |
-1.53/0.136 |
1.66/0.597 |
63.79 |
10.48 |
-1.94/0.061 |
3.28/0.295 |
CGM |
61.61 |
10.47 |
62.97 |
10.55 |
-0.79/0.433 |
|
66.42 |
10.35 |
-2.74/0.010 |
|
HbA1c |
|
|
|
|
|
Nil |
|
|
|
-0.21/0.552 |
Control |
9.45 |
1.34 |
Nil |
Nil |
Nil |
|
9.08 |
1.73 |
1.08/0.289 |
|
CGM |
9.37 |
1.45 |
Nil |
Nil |
Nil |
|
8.79 |
1.76 |
2.51/0.017 |
|
Fasting blood glucose |
|
|
|
|
|
Nil |
|
|
|
0.89/0.958 |
Control |
173.50 |
61.26 |
Nil |
Nil |
Nil |
|
163.72 |
45.40 |
0.54/0.593 |
|
CGM |
167.33 |
66.27 |
Nil |
Nil |
Nil |
|
158.44 |
61.12 |
0.68/0.503 |
|
1Paired t-testï¼›2 Generalized estimating equation (GEE) statistics
Comment 2: The English can be improved.
Response: We have revised the sections of results, discussion, and conclusion on page [6-12], paragraph [multiples], lines [263-392]:
Results section:
“3.2. Comparisons of outcomes between and within groups. At different time points, no significant differences were observed between the CGM and control groups regarding medication adherence, diabetes self-efficacy, diabetes health behaviors, HbA1c (%), and fasting glucose (mg/dl). Within-group analyses further revealed no significant differences in medication adherence scale or fasting blood glucose levels across time points (T0, T1, and T2) for either group.
However, a notable difference was found in diabetes self-efficacy and health behaviors. The control group exhibited a significant decline in self-efficacy scores, dropping from 72.70 to 68.21, representing a 6.18% decrease (p = 0.018). Conversely, the CGM group experienced a slight but non-significant reduction in self-efficacy, with scores declining from 71.82 to 70.61 (a 1.68% decrease, p = 0.647). In terms of diabetes health behaviors; the control group displayed a slight but statistically non-significant increase, with scores rising from 60.52 to 63.79 (a 5.40% increase, p = 0.061). In contrast, the CGM group demonstrated a significant improvement in health behaviors, with scores increasing from 61.61 to 66.42, marking a 7.81% increase (p = 0.010) and an effect size of 0.47, indicating a moderate practical impact.
A similar trend was observed in HbA1c levels. The control group displayed a slight but statistically non-significant reduction in HbA1c from 9.45 to 9.08 (a 3.92% decrease, p = 0.289). Conversely, the CGM group demonstrated a significant reduction in HbA1c levels, declining from 9.37 to 8.79 (a decrease of 6.19%, p = 0.017) and reaching an effect size of 0.83. (Table 2 & Figure 3).”
Discussion section:
“4. Discussion
This preliminary study evaluates the effectiveness of integrating self-regulation education with continuous glucose monitoring (CGM) in diabetes management. Unlike previous studies [13, 14], which primarily focused on clinical outcomes like HbA1c and glucose variation, this study also examines behavioral dimensions, including diabetes self-efficacy and health behaviors. The findings highlight that while no immediate differences were observed between the CGM and control groups, CGM use contributed to significant improvements in diabetes health behaviors, with an effect size approaching a moderate level, and glycemic control, with a large effect size over time. These results underscore the potential of CGM as a valuable tool not only for enhancing clinical outcomes but also for fostering positive behavioral changes in diabetes management.
While the CGM group consistently maintained their diabetes self-efficacy scores throughout the study, a notable decline was observed in the control group at the third time point (T2). Interestingly, these findings contrast with those reported in a systematic review [15], which highlighted a significant and positive impact of diabetes education self-management interventions on enhancing self-efficacy among patients with type 2 diabetes. The challenge of sustaining or enhancing self-efficacy following diabetes education interventions can be multifaceted. One possible explanation for the decline in the control group is an initial overestimation of self-efficacy, a phenomenon observed in previous research on diabetes self-management programs [16]. As participants engaged in the diabetes education program and closely monitored their behaviors over time, they gained a more realistic understanding of their actual capabilities and encountered the inherent challenges of diabetes management. This increased awareness may have contributed to the observed decline in self-efficacy scores among the control group. Additionally, the observed difference between groups suggests that the CGM intervention may have played a role in sustaining self-efficacy levels. Unlike the control group, participants in the CGM group received continuous real-time feedback on glucose fluctuations, allowing them to make immediate behavioral adjustments and observe the direct impact of their actions. This reinforcement mechanism may have enhanced their confidence in managing their condition, helping them sustain their self-efficacy scores despite the complexities of diabetes self-management. In contrast, the control group lacked this immediate feedback loop, which may have contributed to a gradual decline in their perceived ability to manage diabetes effectively.
The discrepancy in health behaviors observed in both groups highlights the varying impact of interventions on diabetes management. The CGM group demonstrated a significant improvement in diabetes health behaviors after three months, whereas the control group exhibited no significant differences across time points. This finding indicates that the integration of CGM with self-regulation education may facilitate the adoption of health-promoting behaviors, potentially reinforcing the application of diabetes management strategies in daily life. The ability to receive real-time glucose feedback may have played a crucial role in fostering behavioral adjustments and supporting long-term adherence to health-promoting practices. These results align with evidence from a review study [17], further emphasizing the value of incorporating CGM as a complementary tool in diabetes self-management interventions. Moreover, the significant increase in health behavior scores observed in the CGM group suggests that participants not only acquired relevant knowledge and skills through the education program but also actively translated them into improved diabetes management practices, highlighting the potential of CGM to bridge the gap between education and sustained behavior change.”
Conclusion section:
“This study underscores the potential benefits of combining Continuous Glucose Monitoring (CGM) with self-regulation education in diabetes management. While no differences were observed between the CGM and control groups in medication adherence, diabetes self-efficacy, diabetes health behaviors, and glycemic control after the intervention, the CGM group demonstrated sustained engagement in diabetes management and significant improvements in both diabetes health behaviors and HbA1c levels over time. These findings highlight the importance of personalized, patient-centered approaches, highlighting the need for ongoing support and reinforcement through self-regulation education. Additionally, the role of the CGM device in providing real-time feedback appears to be a crucial factor in facilitating behavioral adjustments and improving long-term outcomes. Overall, this study provides valuable insights into the advantages of integrating CGM with self-regulation education, advocating for continuous education and individualized care strategies to optimize diabetes self-management and achieve better glycemic control.”
